# Molecular Detection of Venous Thrombosis in Mouse Models Using SPECT/CT

**DOI:** 10.3390/biom12060829

**Published:** 2022-06-13

**Authors:** Annemiek Dickhout, Pieter Van de Vijver, Nicole Bitsch, Stefan J. van Hoof, Stella L. G. D. Thomassen, Steffen Massberg, Peter Timmerman, Frank Verhaegen, Rory R. Koenen, Ingrid Dijkgraaf, Tilman M. Hackeng

**Affiliations:** 1Department of Biochemistry, CARIM—School for Cardiovascular Diseases, Maastricht University, 6229 ER Maastricht, The Netherlands; annemiekdickhout@hotmail.com (A.D.); pieter.vandevijver@gmail.com (P.V.d.V.); s.thomassen@maastrichtuniversity.nl (S.L.G.D.T.); r.koenen@maastrichtuniversity.nl (R.R.K.); i.dijkgraaf@maastrichtuniversity.nl (I.D.); 2Muroidean Facility, CARIM—School for Cardiovascular Diseases, Maastricht University, 6229 ER Maastricht, The Netherlands; nicole.bitsch@maastrichtuniversity.nl; 3Department of Radiation Oncology (MAASTRO), GROW—School for Oncology and Developmental Biology, Maastricht University Medical Center, 6229 ER Maastricht, The Netherlands; stefan.vanhoof@smartscientific.nl; 4Department of Medicine I, University Hospital, LMU Munich, 80336 Munich, Germany; steffen.massberg@med.uni-muenchen.de; 5German Centre for Cardiovascular Research (DZHK), Partner Site Munich Heart Alliance, 10785 Berlin, Germany; 6Pepscan Therapeutics B.V., 8243 RC Lelystad, The Netherlands; p.timmerman@uva.nl; 7Department of Radiology and Nuclear Medicine, Maastricht University Medical Center, 6229 HX Maastricht, The Netherlands; frank.verhaegen@maastro.nl

**Keywords:** thrombosis, fibrin, molecular imaging, SPECT, thrombolysis

## Abstract

The efficacy of thrombolysis is inversely correlated with thrombus age. During early thrombogenesis, activated factor XIII (FXIIIa) cross-links α2-AP to fibrin to protect it from early lysis. This was exploited to develop an α2-AP-based imaging agent to detect early clot formation likely susceptible to thrombolysis treatment. In this study, this imaging probe was improved and validated using ${}^{111}$In SPECT/CT in a mouse thrombosis model. In vitro fluorescent- and ${}^{111}$In-labelled imaging probe-to-fibrin cross-linking assays were performed. Thrombus formation was induced in C57Bl/6 mice by endothelial damage (FeCl${}_{3}$) or by ligation (stenosis) of the infrarenal vena cava (IVC). Two or six hours post-surgery, mice were injected with ${}^{111}$In-DTPA-A16 and ExiTron Nano 12000, and binding of the imaging tracer to thrombi was assessed by SPECT/CT. Subsequently, ex vivo IVCs were subjected to autoradiography and histochemical analysis for platelets and fibrin. Efficient in vitro cross-linking of A16 imaging probe to fibrin was obtained. In vivo IVC thrombosis models yielded stable platelet-rich thrombi with FeCl${}_{3}$ and fibrin and red cell-rich thrombi with stenosis. In the stenosis model, clot formation in the *vena cava* corresponded with a SPECT hotspot using an A16 imaging probe as a molecular tracer. The fibrin-targeting A16 probe showed specific binding to mouse thrombi in in vitro assays and the in vivo DVT model. The use of specific and covalent fibrin-binding probes might enable the clinical non-invasive imaging of early and active thrombosis.

## 1. Introduction

With a globally aging population, the lifetime risk of thrombo-embolic and ischemic diseases is increasing [1,2,3]. Survival rates after thrombo-embolic disease, such as pulmonary embolism (PE) but also ischemic stroke and myocardial infarction, are inversely correlated with time to treatment. Therefore, it is important to diagnose these diseases early after onset. Fibrinolytic therapy (lysis) is the first line of treatment for these ischemic diseases. Still, resistance to lysis increases with thrombus age, while the hazard of severe side effects such as gastrointestinal bleeding or intracerebral haemorrhage remains [4,5,6]. The first hours after thrombus formation is the timeframe in which thrombolytic treatment with tissue plasminogen activator (tPA) is indicated for PE, ischemic stroke and ST-segment elevation myocardial infarction (STEMI), as well as other thrombosis-related off-label indications. Therefore, diagnosing early thrombus formation will aid in selecting patients that will benefit from fibrinolytic therapy.

Current clinical prediction tools are indirect and based on changes in anatomy or function, whereby diagnostic strategies include D-dimer testing, ventilation-perfusion (VQ) scanning, or computed tomography pulmonary angiography (CTPA) [7]. Other diagnostic tools used are ultrasound, X-ray, CT, or MRI, which are all based on structural changes or the cessation of blood flow rather than the molecular composition of thrombi. The development and improvement of molecular imaging techniques are essential to visualise thrombi at an early stage, enable whole-body or multisite imaging, improve diagnostic specificity and sensitivity, and monitor clinical outcomes [7,8,9,10].

Fibrin has been a target of interest in developing thrombus imaging agents, as fibrin deposition plays a central role in both arterial and venous thrombosis [11,12]. Fibrin is minimally present in the circulation under physiological conditions. However, the precursor fibrinogen is present at 4 mg/mL concentrations. Upon activation, fibrin is formed rapidly and is the ultimate target of thrombolytic enzymes used to treat the clinical presentations associated with thromboembolic diseases, making fibrin a suitable target for molecular imaging [12]. Over the last years, various fibrin-targeting probes have been developed for the imaging modalities SPECT, PET, MRI, and optical imaging [9,13,14,15,16,17,18,19,20,21,22]. Most of these peptide-based probes are small and can consequently penetrate easily into thrombi. They are easy to synthesize, are less likely to be immunogenic, and have a rapid clearance from the blood [23]. Several peptidic probes based on α2-antiplasmin (α2-AP) rely on the transglutaminase activity of factor XIIIa (FXIIIa) for [24,25], leading to the covalent cross-linking of the probes to fibrin during early-phase thrombin formation [26,27]. It appeared that α2-AP-based probes not only enable in vitro and in vivo visualisation of thrombi but also the distinction between new and older thrombi, which could facilitate the selection of patients that would benefit most from thrombolytic therapy [25,28].

This proof-of-concept study aimed to develop an optimized α2-AP-based nuclear imaging probe by increasing the hydrophilicity and fibrin-binding potential of our previous bimodal α2-AP-based contrast agent (bi-α2-AP-CA) [25]. Therefore, in the new probe A16, tryptophan-14 was replaced by β-alanine-lysine-lysine and conjugated to diethylenetriamine pentaacetic acid (DTPA) for labelling with indium-111 (${}^{111}$In) or Lissamine rhodamine-B via lysine-13. Substitution of glutamine-3 to an alanine residue resulted in the control probe control-A16 that cannot be coupled to fibrin by FXIIIa.

Here, the performance of this new α2-AP-based probe was first assessed in an in vitro plasma clotting assay using the rhodamine- and ${}^{111}$In-labelled variant. Then, immunohistochemistry compared two different mice deep venous thrombosis (DVT) models for their fibrin composition. The two models include the previously used ferric chloride model and a stenosis model, induced in the infrarenal vena cava (IVC), where endothelial damage by ferric chloride is a model for a more rapid, non-DVT-like platelet-rich thrombosis, and the stenosis model represents DVT [29]. The IVC stenosis model was used to assess the potential of the ${}^{111}$In-labelled tracer in SPECT/CT. Finally, the biodistribution of the nuclear imaging tracer was determined, and ex vivo scans of IVCs were made.

## 2. Materials and Methods

### 2.1. Peptide Synthesis and Radiolabelling

The bimodal peptides bi-α2-AP-CA and control-bi-CA were synthesized using tert-butyloxycarbonyl solid-phase peptide synthesis [30] as described by Miserus et al. [25]. A16 and control-A16 were synthesized and conjugated with either DTPA or Lissamine rhodamine-B (LisB) by Fmoc-based solid-phase peptide synthesis using standard protocols [31]. ${}^{111}$In-A16 and ${}^{111}$In-control-A16 were prepared by adding 50–60 MBq ${}^{111}$InCl${}_{3}$ (Mallinckrodt, Petten B.V., The Netherlands) to 10 μg of DPTA-conjugated A16 (4.6 nmol) or control-A16 (4.7 nmol), dissolved in 200–300 μL of 0.1 M 2-(N-morpholino)ethane sulfonic acid (MES) buffer of pH 5.5. Radiolabelling was performed for 20 min at room temperature. Radiochemical purity was determined by RP-HPLC (LC-20AT, Shimadzu Benelux, ‘s-Hertogenbosch, The Netherlands) using a C18 column (RP-C18 Inertsil ODS-3, 4.6 × 250 mm, 5 μM, Phenomenex, Utrecht, The Netherlands) eluted with a linear gradient of CH${}_{3}$CN (0–100% in 30 min) in H${}_{2}$O containing 0.1% trifluoroacetic acid (TFA; *v/v*) at a flow rate of 1 mL/min. The radioactivity was monitored using an in-line radio detector (Gabi, Raytest GmbH, Straubenhardt, Germany). The radiochemical purity of the preparations used in in vitro and in vivo experiments always exceeded 95%. Radiotracers were used without further purification. The ${}^{111}$In was preferred over other radionuclides (e.g., ${}^{99m}$Tc) because the complex of ${}^{111}$In and DTPA is more stable in plasma [32].

### 2.2. In Vitro Probe Validation

Human blood was collected from healthy volunteers by venipuncture using a vacutainer tube containing trisodium citrate after signing informed consent (Helsinki declaration). Thrombi (50 μL plasma) were allowed to form for 90 min at 37 °C with shaking in the presence of 14 mM CaCl${}_{2}$ and 0.6 nM TF and subsequently incubated with 3 μM bi-α2-AP-CA, control-bi-CA, LisB-A16 or LisB-control-A16. At the indicated time points, the OD570 of the supernatant was measured. Mouse blood was collected through a tail vein puncture, and plasma from 6 mice was pooled. Human and mouse thrombi were allowed to form as described above and subsequently incubated with 100 μL or 16 ng/μL ${}^{111}$In-A16 or ${}^{111}$In-control-A16. After 30, 60 and 180 min, the thrombi were washed twice with 1 mL PBS, and the amount of tracer uptake in the thrombi was calculated using gamma counting (Wallac Wizard, Turku, Finland). Data were expressed as the percentage of tracer uptake in the thrombi to the total amount of tracer added.

### 2.3. Animals

C57BL/6 male mice (8–12 weeks old, Charles River, The Netherlands) were used for all in vivo experiments. Animal experimental procedures were approved by the Institutional Animal Care and Use Committee of Maastricht University (Nr. 2013-076, 10 June 2014). All protocols were carried out in compliance with the Dutch government guidelines and the guidelines from Directive 2010/63/EU of the European Parliament on the protection of animals used for scientific purposes.

### 2.4. Ferric Chloride Thrombosis Model

The ferric chloride thrombosis model was performed analogously as described by Wang and colleagues [33]. Briefly, C57BL/6 male mice were anaesthetized using 3–5% isoflurane (IsoFlow, Zoetis B.V., Rotterdam, The Netherlands) and 0.05 mg/kg fentanyl (Eli Lilly, Indianapolis, IN, USA) subcutaneously (s.c.), and received a median laparotomy. The IVC was exposed, and a piece of filter paper soaked in a 10% FeCl${}_{3}$ solution in distilled water was placed just under the left renal vein for 5 min. After removal and washing with 0.9% NaCl, the incision was closed using a 7-0 prolene suture (Ethicon, Johnson & Johnson Medical N.V., Diegem, Belgium). The mice received 0.05 mg/kg buprenorphine s.c. before waking and at regular intervals until the end of the experiment. Sham surgery involved exposing the IVC and placing a filter paper soaked in sterile water.

### 2.5. Ivc Stenosis Thrombosis Model

The model was performed as described previously by von Brühl and colleagues [11]. Briefly, C57BL/6 male mice were anaesthetized using 3–5% isoflurane and 0.05 mg/kg fentanyl s.c. and received a median laparotomy. The IVC was exposed, and a ligation was placed around the IVC, and a 0.014 inch space holder was placed just below the left renal vein using an 8-0 prolene monofilament suture (Ethicon). The space holder was removed to avoid complete vessel occlusion, and the incision was closed. Mice received 0.05 mg/kg buprenorphine s.c. before waking and at regular intervals until the end of the experiment. Sham surgery involved exposing the IVC and placing a filament without ligation.

### 2.6. Computed Tomography (CT)

Animals received 100 μL ExiTron Nano 12,000 intravenously (i.v.; Viscover, Berlin, Germany) through a tail vein injection and were subsequently anaesthetized using 3–5% isoflurane (Zoetis). Cone beam computer tomography (CBCT) scanning was performed using a micro CT (Xrad 225Cx, Precision X-ray, USA) and accompanying PilotCal software at 80 kVp.

### 2.7. Single-Photon Emission Computed Tomography (SPECT)

Six hours after induction of thrombosis, 13.2 ± 0.4 MBq ${}^{111}$In-A16 (4.1 μg) was administered through a tail vein cannula. Mice were anaesthetized with 3–5% isoflurane and positioned in the SPECT camera (U-SPECT, VECTor, acquisition version 3.7ds, MiLabs, Utrecht, The Netherlands). A whole-body SPECT (4 × 15 min) was performed using a 0.6 mm collimator, after which the animal was placed in the micro CT (Xrad 225Cx, Precision X-ray, Madison, CT, USA). CT imaging was performed as described above. To facilitate co-registration of SPECT/CT images, external markers on the animal beds containing ${}^{111}$In and ExiTron Nano 12,000 were used. Rigid co-registration was manually performed with PMOD image fusion (Bruker, Billerica, MA, USA). After SPECT/CT, animals were dissected, and ex vivo scans were made of the IVCs. Therefore, after gamma counting (as described below), the IVCs were imbedded in 2% agarose gel and scanned for 4 × 1 h.

### 2.8. Biodistribution

After SPECT/CT scanning, animals were euthanized, and major organs and tissues were collected, weighed, and counted in an automated NaI(TI) gamma counter (Wallac Wizard, Turku, Finland). Data are expressed as the percentage of injected dose (ID) per gram of tissue (%ID/g).

### 2.9. Histology

The complete *V. cava* and aorta were fixed for 48 h in 4% formalin, dehydrated and embedded in paraffin. Sequential sections of 5 μM were cut using a sliding microtome. After deparaffinisation and rehydration, sections were stained with haematoxylin and eosin (HE) (Klinipath, Duiven, The Netherlands), or Carstairs’ method for fibrin and platelets (EMS 26381), which stains fibrin (bright red), platelets (grey-blue), collagen (bright blue) and red blood cells (yellow). Images were taken using a light microscope (Leica DM RBE and a DFC425C camera) and analysed using ImageJ (U. S. National Institutes of Health, Bethesda, MD, USA).

## 3. Results

### 3.1. Peptide Synthesis

Structure formulas of bi-α2-AP-CA, DTPA-A16 (Ac-GNQEQVSPLTLLK1-K(DTPA)K-NH${}_{2}$, 1 = βAla), and their Q3A counterparts, which served as controls, are given in Figure 1A,B, respectively. Matrix-assisted laser desorption ionization mass spectrometry of bi-α2-AP-CA (Figure 1C) and A16 (Figure 1D) showed a mass of 2832.86 and 2170.89 Da, which corresponded to the calculated masses of 2832.92 and 2170.15 Da, respectively. In the mass spectrum of bi-α2-AP-CA, a small peak is visible at 2169.85 Da, representing the loss of maleimide-DTPA. The mass spectrum of A16 demonstrated a small extra peak at 2224.80 Da, representing chelated Fe${}^{3+}$. The structure of the red fluorescent probe LisB-A16 (Ac-GNQEQVSPLTLLK1-K(LisB)K-NH${}_{2}$, 1 = βAla) for initial in vitro testing is given in Appendix A.

### 3.2. In Vitro Probe Incorporation

LisB-A16 was compared with bi-α2-AP-CA. Human plasma was allowed to clot for 90 min at 37 °C, after which a 3 μM probe was added. The decrease in probe left in the solution was measured by colourimetry and represented probe incorporation into the human thrombi (Figure 2A). The absorption at 570 nm remained close to 100% for the Q3A control probes (dotted lines) and reduced to approx. 67% for bi-α2-AP-CA (solid red line) compared to 53% for LisB-A16 (solid green line), demonstrating an absence of incorporation of the controls and suggesting more effective incorporation of LisB-A16 than bi-α2-AP-CA. At a higher probe concentration (15 μM), the remaining probe in solution was 84% (bi-α2-AP-CA) and 61% (LisB-A16; Appendix A). Representative images of thrombi incubated with LisB-control-A16 and LisB-A16 clearly showed that LisB-A16 accumulated in the fibrin clot in contrast to LisB-control-A16 (Figure 2B).

Subsequently, the incorporation of ${}^{111}$In-labelled A16 and control-A16 were tested in vitro both in human and mouse thrombi to verify the uptake of tracer into mouse thrombi before evaluating the probes in a mouse DVT model. The RP-HPLC elution profiles of ${}^{111}$In-A16 and ${}^{111}$In-control-A16 showed a single peak for both compounds with elution times of 12.7 and 14.9 min for ${}^{111}$In-A16 and ${}^{111}$In-control-A16, respectively (Appendix A). ${}^{111}$In-A16 was added to human and mouse plasma that had been allowed to clot for 90 min. After 0.5, 1 or 3 h, thrombi were washed, and the amount of tracer uptake was measured using gamma counting. After 3 h, approximately 50% of the tracer was taken up by both the mouse (approx. 55%) and human (approx. 45%) thrombi, while about 25% of control-A16 was incorporated (Figure 3).

### 3.3. Validation of Mouse DVT Models

To test the new probe in vivo, two different DVT C57BL/6 mouse models were used. The rapid IVC FeCl${}_{3}$ model was compared to the IVC stenosis model because the cellular composition and fibrin content are different in the resulting thrombi [11,34]. Prior to testing the tracer, the models were validated at different time points using CBCT scanning with a single i.v. bolus injection of the contrast agent ExiTron Nano 12000. Representative images are shown in Figure 4, showing an intact *V. cava* in the mouse receiving sham surgery (A), a lack of contrast on the apical side of the *V. cava* in the mouse that received FeCl${}_{3}$ treatment, indicating thrombus formation (B), and a clear negative contrast throughout the *V. cava* and the location of ligation, indicating an occlusive thrombus in the IVC stenosis model (C).

After CBCT scanning, mice were dissected in order to validate the formation of a thrombus and for the preparation of histologic specimens. Using FeCl${}_{3}$, all of the mice experienced rapid thrombosis (*n* = 2, 2 h, *n* = 2, 6 h, *n* = 4, 24 h), with highly variable size, as has been described before [34]. Due to this variable size, only 3 out of 8 thrombi were visible in CBCT scanning. Since the indication of thrombosis is based on the absence of i.v. contrast, small or superficial thrombi were not detectable. In the IVC stenosis model, 13 out of 15 mice developed thrombosis (*n* = 5/6 6 h, *n* = 4/4 24 h, *n* = 4/5 48 h), with lower variability in thrombus size than reported [34]. Thrombus specimens were stained using haematoxylin/eosin staining (Figure 5A,C) and Carstairs’ method for fibrin and platelets (Figure 5B,D,E). FeCl${}_{3}$-induced thrombi were rich in platelets (Figure 5A,B), whereas the IVC stenosis showed a higher content of fibrin and red blood cells (Figure 5C,D). Sagittal sectioning showed a clear, typical layered pattern of platelets, fibrin and red blood cells of the thrombus (Figure 5E), as observed previously [11].

### 3.4. In Vivo Thrombus Imaging in Vena Cava

As the imaging probe is based on α2-AP and will bind covalently to fibrin, we hypothesized that the tracer uptake would be higher in the IVC stenosis model since the thrombi contain more fibrin than in the FeCl${}_{3}$ model. In addition, a previous study demonstrated that stasis-induced thrombi were rich in α2-AP in mice and that α2-AP was a strong determinant of thrombus size, indicating an important role of the fibrinolytic pathway in mouse thrombosis [35]. As proof of concept, we imaged uptake of ${}^{111}$In-A16 in a C57BL/6 mouse six hours after IVC ligation and in sham-operated mice. Therefore, mice were injected with 13.2 ± 0.4 MBq tracer (200 μg/kg body weight) and immediately scanned using SPECT/CT. Figure 6A–D shows representative sections of sagittal (A) and coronal (B) views and a 3D model (C) of a mouse. High uptake was visible in the kidneys and bladder, indicating predominant renal clearance of the tracer. A small droplet of urine is seen at the base of the tail. Asterisks (*) indicate the location of the IVC ligation, where a hyperintense signal indicates uptake of the tracer at the location of the thrombus. Biodistribution studies of the tracer showed high uptake in *V. cava* and kidneys, indicating uptake of the tracer in the thrombus and renal clearance (Figure 6E). Ex vivo SPECT scans of paraffin-embedded IVCs clearly showed high ${}^{111}$In-A16 uptake in the IVC of a mouse that received ligation (Figure 6E, top) compared to a mouse that received sham surgery (bottom), indicating that a thrombus is present in the IVC for tracer uptake. Figure 6F shows pictures of the embedded thrombi used for ex vivo scanning. However, in most of our experiments, including mice treated with FeCl${}_{3}$, whole-body SPECT did not show uptake of the tracer in vivo, but ex vivo scans of the IVCs with thrombi did reveal uptake of the tracer (Appendix A).

## 4. Discussion

Since cardiovascular disease (CVD) causes major morbidity and mortality worldwide, there is a constant need for the improvement of diagnostic procedures. In this study, we designed and developed an optimized α2-AP-based molecular imaging probe, A16, which is covalently bound to fibrin by FXIIIa during thrombosis. This imaging agent enables molecular imaging of active and early thrombotic processes and is therefore attractive for clinical translation. In in vitro human plasma clot formation assay LisB-labelled A16 demonstrated higher uptake in these plasma clots than bi-α2AP-CA, which could be explained by increased hydrophilicity due to the replacement of tryptophan-14 in bi-α2AP-CA by the tripeptide sequence β-alanine-lysine-lysine in A16. However, the overall increase in hydrophilicity cannot be determined because LisB-A16 lacks the hydrophilic DTPA moiety present in the bimodal bi-α2AP-CA.

This study aimed to further develop our optical imaging tracer into a radionuclide-based tracer, enabling visualization of lesions deeper in tissues and organs. Therefore, DTPA-conjugated A16 was synthesized and radiolabelled with ${}^{111}$In. In in vitro assays, ${}^{111}$In-A16 showed faster uptake in mice compared to human thrombi. This observation could be explained by the overall higher activity of coagulation enzymes present in mouse plasma, as reported previously [36]. The role of Q3 in the covalent coupling of A16 to fibrin was confirmed by a significantly lower uptake of ${}^{111}$In-control-A16 in thrombi of both species. Before investigating the in vivo thrombus-targeting potential of the newly developed radiotracer, two mouse DVT models were compared in terms of cellular and fibrin composition by CT and IHC. The FeCl${}_{3}$ endothelial injury model resulted in more platelet-rich thrombi, whereas the stenosis model showed a typically layered pattern of white and red thrombi, which is in line with previous research [11,34,37,38]. Thrombi obtained from the stenosis model showed a higher fibrin content, and the thrombus size was more stable than the frequently used FeCl${}_{3}$ model. Furthermore, this model showed structural features similar to human venous thrombosis [11]. As the stenosis model is more in line with the clinical situation and the thrombi formed were demonstrated to possess a higher fibrin content, we decided to evaluate ${}^{111}$In-A16 in the stenosis model only. ${}^{111}$In-A16 showed high uptake in the *V. cava* with thrombus and cleared rapidly from the blood via the kidneys. Uptake in non-target organs and tissues such as muscle was low, demonstrating the specificity of ${}^{111}$In-A16.

The advantage of radionuclide-based imaging techniques like PET and SPECT is their sensitivity to nano- and even picomolar tracer concentrations. Indeed, the dose used in the SPECT experiments (0.1 μmol/kg) was over 40 times lower than the dose previously used for the gadolinium-labelled MRI α2-AP based probe (5 μmol/kg) [25], reducing the potential for pharmacological effects and toxicity [9]. Although ex vivo scanning of the *V. cava* showed uptake of ${}^{111}$In-A16 in mouse thrombi, the in vivo thrombus could not be visualized in all SPECT experiments. Possible explanations include the limited spatial resolution of the μSPECT used. In the setup used, with a 0.6-mm pinhole collimator, submillimetre resolution was the maximum achievable resolution [39]. Performance of the SPECT is dependent on multiple factors, including reconstruction algorithms and the type of isotope used. Due to the relatively high energy of ${}^{111}$In-emitted photons leading to increased collimator scatter, ${}^{111}$In is among the more challenging isotopes to obtain SPECT images [40,41], yet ${}^{111}$In-DTPA shows robust stability in plasma [32]. Furthermore, thrombi were induced directly between the highly radioactive kidneys and bladder, which all together could lead to failure to detect the thrombus in vivo while uptake of the tracer was clearly observed ex vivo. In addition, a possible disadvantage of the stenosis model is that injected compounds are unable to reach the thrombus once occlusion of the vein has occurred, which may affect signal intensity [38]. As ${}^{111}$In-A16 is covalently bound to fibrin during thrombus formation, and the half-life of ${}^{111}$In is 2.8 days, it would be interesting to see if tracer imaging is improved if A16 is injected after thrombus induction and imaged several hours or one day later, even though this setup steers away from the clinical setting. To overcome scattering issues from the bladder and kidneys, imaging of thrombus formation in the femoral or saphenous vein would be more conducive [42]. Since an important property of the probe is its covalent binding to the thrombus, it is of great importance to test the pharmacodynamic interaction with thrombolytic agents, e.g., tissue plasminogen activator. In addition, dose optimization is crucial during future follow-up studies.

It is important to note that the results obtained in mouse models may not be optimally translatable to humans. Despite the wide range of choices of mouse models for thrombosis, every model only covers a particular pathophysiologic aspect of clinical thrombosis in humans [34,38]. This is also highlighted in this study, where FeCl${}_{3}$-induced thrombi were found to be rich in platelets, and the stenosis-induced thrombi rather contained fibrin. Thus, the choice of mouse model determines the outcome, which should be kept in mind when drawing conclusions about clinical implications. Since mice have a larger metabolic rate [38] and a higher activity of anticoagulant factors [36], thrombin generation and fibrinolysis will likely be different compared to humans. In this study, the choice was made for the stenosis-induced thrombosis model because the thrombi obtained are fibrin-rich. Moreover, the fibrinolytic system was shown to be important in stasis-induced thrombosis in a recent mouse study [35]. Genetic deletion of α2-AP resulted in a significant reduction of thrombus size, indicating a prominent role of fibrinolysis and α2-AP in stasis-induced thrombosis. This is also reflected by the detection of ${}^{111}$In-DTPA-A16 incorporation in the mouse thrombus. In human acute thrombotic events, aberrant blood flow may not always play a decisive role in disease etiology. While abnormal flow might be a prominent contributor to deep vein thrombosis, acute ischemic stroke might rather be caused by cardiogenic emboli as a complication of atrial fibrillation or driven by massive platelet aggregation due to the rupture of carotid lesions. Thus, it is not known whether A16 binds to all thrombi, and it might be worthwhile to target activated platelets for molecular imaging in combination with fibrin-binding probes to cover a broader spectrum of thrombotic manifestations. Indeed, imaging agents directed against the high-affinity conformation of the fibrinogen-binding integrin $\alpha_{2b}\beta_{3}$ are being developed for in vivo detection of platelet aggregation [10,43]. Taken together, it remains to be determined whether A16 can be used for diagnosis in patients.

In conclusion, this study showed a viable new SPECT imaging tracer targeting active thrombus formation through FXIIIa activity. Using a single approach, it harbours the potential to diagnose thrombosis and might be implemented to predict the outcome of thrombolytic therapy. This proof-of-concept study is a step toward the development of an α2-AP-based imaging probe that opens up the potential for clinical imaging of active thrombotic processes.

## Figures and Tables

**Figure 1 biomolecules-12-00829-f001:**
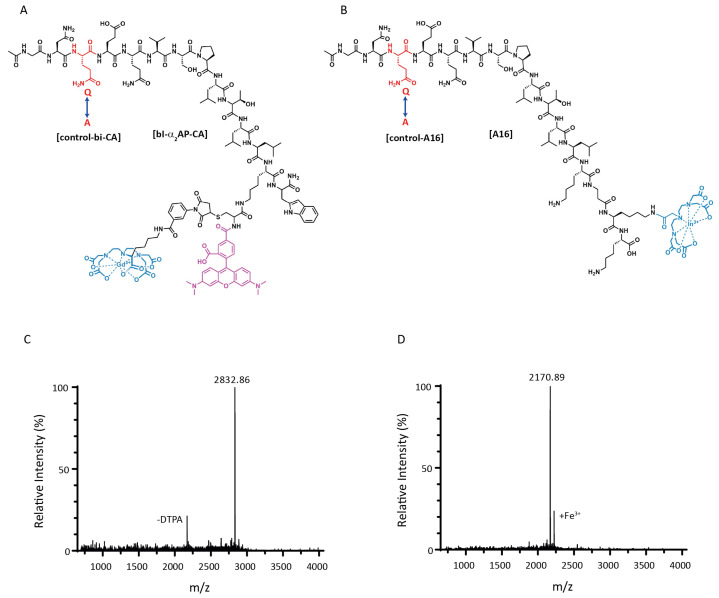
Schematic representations of the previously developed bi-α2-AP-CA (**A**, modified from [25]), and the optimized probe A16 (**B**), which is conjugated to the chelator DTPA (blue), enabling labelling with ${}^{111}$In. Substitution of glutamine to alanine (Q3→A3) leads to a control probe (control-bi-CA or control-A16, respectively), which does not bind to fibrin (**C**,**D**). Mass spectra of bi-α2-AP-CA and A16, respectively, showing a mass of 2832.86 and 2170.89 (calculated masses: 2832.92 and 2170.15).

**Figure 2 biomolecules-12-00829-f002:**
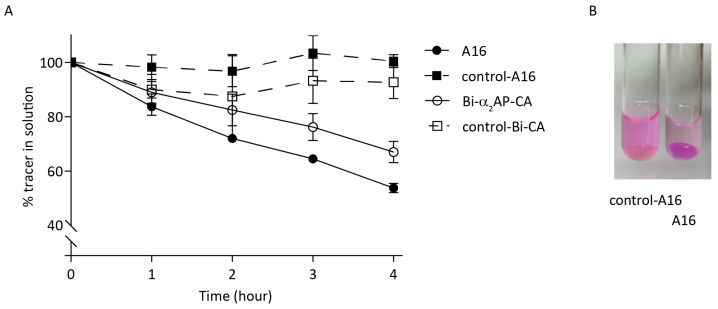
In vitro probe depletion. Human plasma was allowed to form thrombi in vitro at 37 °C for 90 min. (**A**) Rhodamine-labelled probes (3 μM) were added to human thrombi. Solid lines = peptides (Q3), dashed lines = control peptides (Q3→A3), parent peptide (□,○), newly developed A16 (▪,●). OD (570 nm) in solution was measured after 1, 2, 3 and 4 h as a measure of probe incorporation in thrombi. Dots represent mean ± SD, *n* = 4. (**B**) Representative uptake of LisB-A16 vs control-LisB-A16 in thrombi.

**Figure 3 biomolecules-12-00829-f003:**
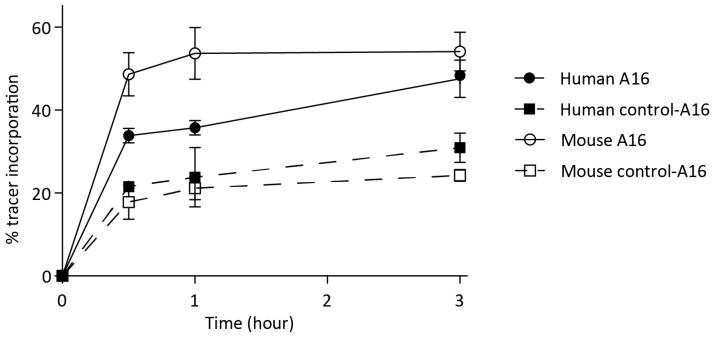
${}^{111}$In-labelled tracers (A16—solid line, control-A16—dotted line) were added to human (▪,●) and mouse (□,○) thrombi. At indicated time points, thrombi were washed, and the amount of tracer uptake was measured in a gamma counter.

**Figure 4 biomolecules-12-00829-f004:**
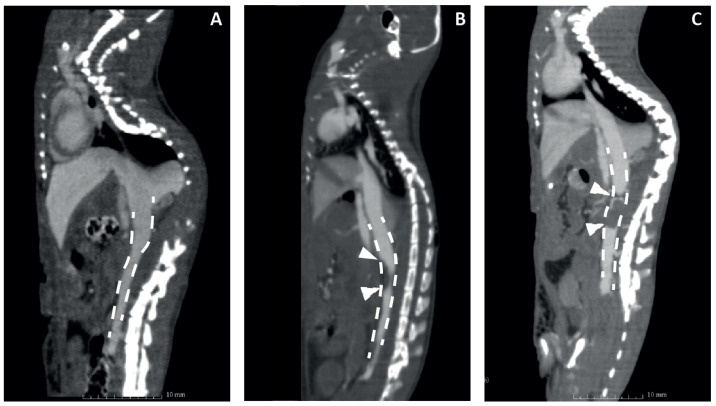
Evaluation of mouse DVT models by contrast-enhanced CT in a sham-operated mouse (**A**), 6 h after endothelial damage with FeCl${}_{3}$ (**B**) and 24 h after flow restriction by partial ligation (**C**). The dashed line shows the IVC in the abdominal cavity. Arrowheads point to lack of intravenous contrast, indicating thrombus formation.

**Figure 5 biomolecules-12-00829-f005:**
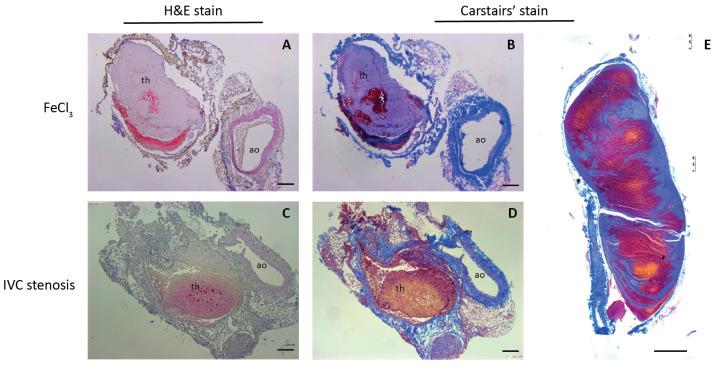
Representative images of transversal slices (**A**–**D**) or sagittal slices (**E**) through thrombi (th) and aorta (ao) induced by FeCl${}_{3}$ (**A**,**B**) or IVC ligation (**C**–**E**). Slices were stained with H&E (**A**,**C**) or Carstairs’ method for fibrin and platelets (**B**,**D**,**E**), which stains fibrin (bright red), platelets (grey-blue), collagen (bright blue) and red blood cells (yellow). Scale bar = 100 μM.

**Figure 6 biomolecules-12-00829-f006:**
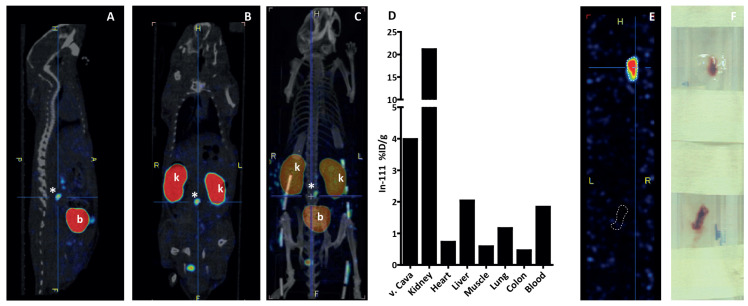
Representative SPECT/CT overlays of sagittal (**A**) and coronal (**B**) views, and a 3D model (**C**) of a mouse 6 hr after IVC ligation, injected with ${}^{111}$In-A16. High uptake of the tracer is seen in kidneys (k), bladder (b) and thrombus (*). (**D**) Accompanying biodistribution of this mouse showing high uptake in the *V. cava* and kidneys, expressed as percentage injected dose per gram tissue (%ID/g). (**E**) Ex vivo SPECT scan of the thrombus (top dotted line). The lower dotted line outlines the IVC of a sham-surgery mouse injected with ${}^{111}$In-A16. (**F**) Light image of embedded thrombi.

## Data Availability

Source data can be obtained from the corresponding author on reasonable request.

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
