# Peer review of "Molecular Detection of Venous Thrombosis in Mouse Models Using SPECT/CT"

_biomolecules, 2022, doi:10.3390/biom12060829_

Round 1

Reviewer 1 Report

The manuscript “Molecular detection of venous thrombosis in mouse models using SPECT/CT” is very well written and the reviewer suggests a minor revision.

My concerns:

The reviewer suggests to discuss the rational of radioisotope selection (why they have used In- versus e.g. Tc-99m or I-123) for this µSPECT study.

The authors shall also discuss/compare the potential and link of α2-antiplasmin and integrin αIIbβ3 imaging.

Unfortunately, the chemical characterization includes only MLDI-MS and the NMR results are missing.

There are few typo mistakes within text and some chemical formula such as CH3CN  (Line 84) and H2O (Line 86).

Reviewer 2 Report

The authors developed the new imaging probe to detect clot formation. They suggested that the probe was developed to detect the a2-antiplasmin molecule and this probe would be useful to find the active and early thrombotic processes. The approach and the data are interesting. However, there are some points to revise. Th detailed information was shown below.

1. Some experiments were performed to mice, especially for the experiments of figure 4, 5, and 6. The authors compared the % tracer incorporation between Human and Mouse thrombin in Figure 3, the coagulation and fibrinolysis cascades and the reaction in mice are not same as human. Although the authors suggested this approach would be useful for the clinical non-invasive imaging, there is not enough evidence for that. Please describe the limitation for that with the difference between human and mice. 

2. Many factors are related to the thrombosis mechanism and the diseases and it is important to consider Virchow’s triad for the thrombosis mechanism. The triad means the three factors related to thrombosis are intravascular vessel wall damage, stasis of flow, and the presence of a hypercoagulable state. When the authors describe the limitation about the difference human and mice, please add the consideration of stasis of flow as the one of limitation because it is considered that stasis of flow is related to the thrombosis mechanism of DVT and PE.

3. The authors suggested that the method could potentially diagnose thrombosis. I understand that this method has the potential to detect clot in early phase and predict the outcome of thrombolysis therapy. However, the data is limited to suggest that as the conclusion. Please explain and describe the prospect based on the current data in the discussion section with the limitation shown above. 

4. One page 3, line 81, RP-HLPC => RP-HPLC
